# Towards Robust and Efficient Contrastive Textual Representation Learning

## Abstract

There has been growing interest in representation learning for text data, based on theoretical arguments and empirical evidence. One important direction involves leveraging contrastive learning to improve learned representations. We propose an application of contrastive learning for intermediate textual feature pairs, to explicitly encourage the model to learn more distinguishable representations. To overcome the learner's degeneracy due to vanishing contrasting signals, we impose Wasserstein constraints on the critic via spectral regularization. Finally, to moderate such an objective from overly regularized training and to enhance learning efficiency, with theoretical justification, we further leverage an active negative-sample-selection procedure to only use high-quality contrast examples. We evaluate the proposed method over a wide range of natural language processing applications, from the perspectives of both supervised and unsupervised learning. Empirical results show consistent improvement over baselines.

## 1 Introduction

Representation learning is one of the pivotal topics in natural language processing (NLP), in both supervised and unsupervised settings. It has been widely recognized that some forms of "general representation" exist beyond specific applications (Oord et al., 2018). To extract such generalizable features, unsupervised representation models are generally pretrained on large-scale text corpora (*e.g.*, BERT (Devlin et al., 2018; Liu et al., 2019; Clark et al., 2020; Lagler et al., 2013)) to avoid data bias. In supervised learning, models are typically built on top of these pre-trained representations and further fine-tuned on downstream tasks. Representation learning greatly expedites model deployment and meanwhile yields performance gains.

There has been growing interest in exploiting contrastive learning (CL) techniques to refine context representations in NLP (Mikolov et al., 2013a;b). These techniques aim to avoid representation collapse for downstream tasks, *i.e.*, getting similar output sentences with different input in conditional generation tasks (Dai & Lin, 2017). Intuitively, these methods carefully engineer features from crafted ("negative") examples to contrast against the features from real ("positive") examples. A feature encoder can then enhance its representation power by characterizing input texts at a finer granularity. Efforts have been made to empirically investigate and theoretically understand the effectiveness of CL in NLP, including noise contrastive estimation (NCE) of word embeddings (Mikolov et al., 2013b) and probabilistic machine translation (Vaswani et al., 2013) with theoretical developments (Gutmann & Hyvärinen, 2010). More recently, InfoNCE (Oord et al., 2018) further links the CL to the optimization of mutual information, which inspired a series of practical followup works (Tian et al., 2020; Hjelm et al., 2019a; He et al., 2020; Chen et al., 2020).

Despite the significant empirical success of CL, there are still many open challenges in its application to NLP, including ($i$) the propagation of stable contrastive signals. An unregularized critic function in CL can suffer from unstable training and gradient vanishing issues, especially in NLP tasks due to the discrete nature of text. The inherent differences between positive and negative textual features make those examples easily distinguished, resulting in a weak learning signal in contrastive schemes (Arora et al., 2019). ($ii$) Empirical evidence (Wu et al., 2017) shows that it is crucial to compare each positive example with adequate negative examples. However, recent works suggest using abundant negative examples, which are not akin to the positive examples, which can result in sub-optimal results and unstable training with additional computational overhead (Ozair et al., 2019; McAllester & Stratos, 2020).

In this paper, we propose two methods to mitigate the above issues. In order to stabilize the training and enhance the model's generalization ability, we propose to use the Wasserstein dependency measure (Ozair et al., 2019) as a substitute for the Kullback-Leibler (KL) measure in the vanilla CL objective. We further actively select $K$ high-quality negative samples to contrast with each positive sample under the current learned representations. These supply the training procedure with necessarily large and non-trivial contrastive samples, encouraging the representation network to generate more distinguishable features. Notably, our approach also significantly alleviates the computational burden of massive features compared with previous works (Tian et al., 2020; Hjelm et al., 2019b).

**Contributions**: ($i$) We propose a Wasserstein-regularized critic to stabilize training in a generic CL framework for learning better textual representations. ($ii$) We further employ an active negative-sample selection method to find high-quality contrastive samples, thus reducing the gradient noise and mitigating the computation concerns. ($iii$) We empirically verify the effectiveness of our approach under various NLP tasks, including variational text generation (Bowman et al., 2016), natural language understanding tasks on GLUE with supervised and semi-supervised setups (Wang et al., 2018), and image-text retrieval (Lee et al., 2018).

## 2 BACKGROUND

### 2.1 NOISE CONTRASTIVE ESTIMATION

Our formulation is inspired by *Noise Contrastive Estimation (NCE)* (Gutmann & Hyvärinen, 2010), which was originally introduced for unnormalized density estimation, where the partition functions is intractable. To estimate a parametric distribution $p$, which we refer to as our *target* distribution, NCE leverages not only the observed samples $A = (\boldsymbol{a}_1, \boldsymbol{a}_2, ..., \boldsymbol{a}_{n_1})$ (positive samples), but also the samples drawn from a *reference* distribution $q$, denoted as $B = (\boldsymbol{b}_1, \boldsymbol{b}_2, ..., \boldsymbol{b}_{n_2})$ (negative samples). Instead of estimating $p$ directly, the density ratio $p/q$ is estimated by training a critic between samples from $A$ and $B$.

Specifically, let $Z = (\boldsymbol{z}_1, ..., \boldsymbol{z}_{n_1+n_2})$ denote the union of $A$ and $B$. A binary class label $C_t$ is assigned to each $\boldsymbol{z}_t$, where $C_t = 1$ if $\boldsymbol{z}_t \in A$ and $C_t = 0$ otherwise.

The label probability is therefore

$$P(C = 1|\boldsymbol{z}) = \frac{p(\boldsymbol{z})}{p(\boldsymbol{z}) + \gamma q(\boldsymbol{z})}, \qquad P(C = 0|\boldsymbol{z}) = \frac{\gamma q(\boldsymbol{z})}{p(\boldsymbol{z}) + \gamma q(\boldsymbol{z})}, \tag{1}$$

where $\gamma = \frac{n_2}{n_1}$ is a balancing hyperparameter accounting for the difference in number of samples between $A$ and $B$.

In practice, we do not know the analytic form of $p$; therefore, a classifier $g : \boldsymbol{z} \mapsto [0, 1]$ to estimate $p(C = 1|\boldsymbol{z})$ is trained. To get an estimation of the critic function $g$, NCE maximizes the log likelihood of the data for a binary classification task:

$$\mathcal{L}(A, B) = \sum_{t=1}^{n_1} \log[g(\boldsymbol{a}_t)] + \sum_{t=1}^{n_2} \log[1 - g(\boldsymbol{b}_t)]. \tag{2}$$

### 2.2 CONTRASTIVE TEXTUAL REPRESENTATION LEARNING AND ITS CHALLENGES

Let $\{\boldsymbol{w}_i\}_{i=1}^n$ be the observed text instances. We are interested in finding a *vector representation $\boldsymbol{u}$* of the text $\boldsymbol{w}$, *i.e.*, via an encoder $\boldsymbol{u} = \text{Enc}(\boldsymbol{w})$, that can be repurposed for downstream tasks. A positive pair refers to paired instances $\boldsymbol{a}_i = (\boldsymbol{u}_i, \boldsymbol{v}_i)$ associated with $\boldsymbol{w}_i$, where we are mostly interested in learning $\boldsymbol{u}$; $\boldsymbol{v}$ is a feature at a different representation level. In unsupervised scenarios, $\boldsymbol{v}_i$ can be the feature representation at the layer next to the input text $\boldsymbol{w}_i$. In supervised scenarios, $\boldsymbol{v}_i$ can be either the feature representation layer immediately after $\boldsymbol{w}_i$ or immediately before the label $\boldsymbol{y}_i$ that corresponds to the input $\boldsymbol{w}_i$. We will use $\pi(\boldsymbol{u}, \boldsymbol{v})$ to denote the joint distribution of the positive pairs, with $\pi_u(\boldsymbol{u})$ and $\pi_v(\boldsymbol{v})$ for the respective marginals.

Contrastive learning follows the principle of "learning by comparison." Specifically, one designs a negative sample distribution $\tau(\boldsymbol{u}', \boldsymbol{v}')$, and attempts to distinguish samples from $\pi(\boldsymbol{u}, \boldsymbol{v})$ and $\tau(\boldsymbol{u}', \boldsymbol{v}')$ with a critic function $g(\boldsymbol{u}, \boldsymbol{v})$. The heuristic is that, using samples from $\tau$ as references (*i.e.*, to contrast against), the learner is advantaged to capture important properties that could have been

otherwise missed (Hjelm et al., 2019a; Oord et al., 2018). A popular choice of $\tau$ is the product of marginals, *i.e.*, $\tau \leftarrow \pi_0(\boldsymbol{u}', \boldsymbol{v}') = \pi_u(\boldsymbol{u}')\pi_v(\boldsymbol{v}')$ where $(\boldsymbol{u}, \boldsymbol{v})$ are independent of each other, so that $\boldsymbol{b}_i = (\boldsymbol{u}_i', \boldsymbol{v}_i') \sim \pi_0$. Inputting the new $\boldsymbol{a}_i$ and $\boldsymbol{b}_i$ to (2), we obtain the new CL loss:

$$\mathcal{L}_{\text{NCE}} = \mathbb{E}_{\boldsymbol{u}, \boldsymbol{v} \sim \pi}[\log g(\boldsymbol{u}, \boldsymbol{v})] + \gamma \mathbb{E}_{\boldsymbol{u}', \boldsymbol{v}' \sim \pi_0}[\log(1 - g(\boldsymbol{u}', \boldsymbol{v}'))] . \tag{3}$$

Note that when $g$ is trained to optimality $g^*(\boldsymbol{u}, \boldsymbol{v}) = p(C = 1 | \boldsymbol{u}, \boldsymbol{v})$ under $\pi_0$, it establishes a lower bound of mutual information (MI) between $\boldsymbol{u}$ and $\boldsymbol{v}$ for the positive distribution (Tian et al., 2020; Neyshabur et al., 2018):

$$\text{MI}(\pi_{\boldsymbol{u}}, \pi_{\boldsymbol{v}}) = \text{KL}(\pi(\boldsymbol{u}, \boldsymbol{v}) || \pi_u(\boldsymbol{u})\pi_v(\boldsymbol{v})) \geq \mathbb{E}_{\boldsymbol{u}, \boldsymbol{v} \sim \pi}[\log g^*(\boldsymbol{u}, \boldsymbol{v})] + \log \gamma . \tag{4}$$

However, there are three concerns regarding why directly applying equation 3 might not be good in practice for learning the contrastive representation of the input text $\boldsymbol{w}$.

- *Robustness.* The first issue concerns the MI's strong sensitivity to small differences in data samples (Ozair et al., 2019; Tschannen et al., 2020). By definition in equation 4, mutual information is a KL divergence. It is well known that the KL divergence is not a metric-aware divergence measure, which implies a minor difference in representation can induce drastic changes in the mutual information, as a special case of KL. Consequently, the learned $g$ could be numerically unstable (Ozair et al., 2019), which makes the learned representations less robust and does not generalize well to downstream tasks, especially when features come from text (Chen et al., 2018).

- *Weak/vanishing contrastive signal.* With a poor initialization or a poor choice of negative samples, the MI will vanish as the $\pi(\boldsymbol{u}, \boldsymbol{v})$ and $\pi_{\boldsymbol{u}}(\boldsymbol{u})\pi_{\boldsymbol{v}}(\boldsymbol{v})$ become far apart, delivering a faint and non-smooth gradient for training. In an extreme case, the support for $\pi(\boldsymbol{u}, \boldsymbol{v})$ and $\pi_{\boldsymbol{u}}(\boldsymbol{u})\pi_{\boldsymbol{v}}(\boldsymbol{v})$ do not overlap, and the MI and the gradient will vanish to zero (Arjovsky et al., 2017).

- *Negative-sample selection strategy.* Learning MI is generally considered sample inefficient. This point can be corroborated from several perspectives, ranging from theoretical arguments to practical considerations. To confidently estimate a lower bound to the MI, one would need a sample size exponential to the mutual information (*i.e.*, $N \geq \exp(I_\pi(\boldsymbol{u}, \boldsymbol{v}))$) (Ozair et al., 2019; McAllester & Stratos, 2018). Also, both theoretical prediction and empirical evidence suggest a large ratio $\gamma$ is needed for good performance (Tian et al., 2020; Hjelm et al., 2019a), imposing potential computational concerns for large training datasets. On the other hand, some studies report a large $\gamma$ can instead deteriorate model performance (Tschannen et al., 2020; Arora et al., 2019). Such a large $\gamma$ is also believed to be problematic especially when a strong association is expected between $\boldsymbol{u}$ and $\boldsymbol{v}$. In that case, the majority of negative samples are so different from positive samples that the comparisons do not lend effective learning signals, but instead randomly drift the training (Gutmann & Hyvärinen, 2010).

## 3 METHOD

### 3.1 MODEL OVERVIEW

We consider two remedies to mitigate the three issues mentioned in Section 2.2, ($i$) Regarding the *robustness* and the *gradient-vanishing* issues, we switch from the MI-based NCE to a Wasserstein-based NCE, by imposing a Wasserstein constraint to the critic function $g$. The Wasserstein distance yields a continuous discrepancy measure over two distributions, even when they have no overlapping support, and suffers less from the issue of numerical instability (Neyshabur et al., 2018). ($ii$) Regarding the *issue of negative samples*, we propose an active negative-sample-selection strategy, to dynamically select the most challenging negative examples on-the-fly. This strategy smooths the learning signal (Wu et al., 2017), effectively improving the CL, meanwhile significantly reducing the computational overhead.

To this end, we propose RECLAIM (RElaxed Contrastive Learning with ActIve Memory selection) as a robust and efficient CL framework. Our learning framework is illustrated in Figure 1. Details are explained in the following sections.

### 3.2 WASSERSTEIN CONSTRAINED CRITIC

In previous work, the critic $g$ is usually chosen to be a naive feed-forward neural network (Tian et al., 2020). However, as discussed in Section 2.2, such a choice of critic function typically leads to a KL-based objective, which suffers from instability and vanishing-gradient issues. Inspired by (Ozair et al., 2019), we replace the KL divergence in equation 4 with a Wasserstein distance. Specifically,

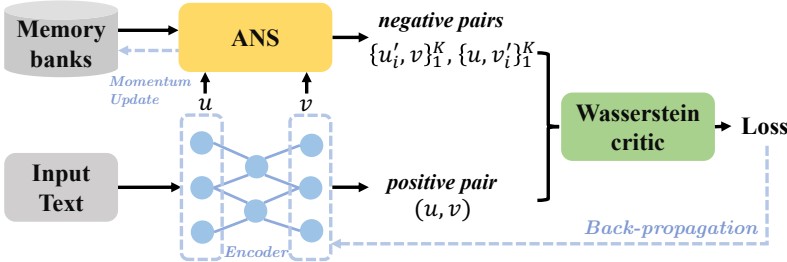

Figure 1: Illustration of our RECLAIM learning framework. An active negative-sample selection (ANS) module is applied for selecting challenging examples. We also relax the critic with the Wasserstein constraint.

we ensure the 1-Lipschitz constraint on the critic function $g$ (Arjovsky et al., 2017). A function $f$ is said to be $L$-Lipschitz if $|f(x) - f(y)| \le L\|x - y\|$, *i.e.*, the difference in function outputs is controlled by the discrepancy in the inputs.

Instead of using a gradient penalty as in (Gulrajani et al., 2017; Ozair et al., 2019), we employ the spectral normalization (SN) (Miyato et al., 2018). We use the SN because it is efficient and also stable. It provides a more strict 1-Lipschitz constraint than the gradient penalty. Specifically, SN controls the Lipschitz constant of the critic function, by constraining each layer $g_l$ in $g$. Formally, it can be formulated as $\|g_l\|_{\text{Lip}} = \sup_x \delta(\nabla(g_l(x)))$, where $\delta(\cdot)$ is the spectral norm, *i.e.*, the largest singular value of the input. For an affine transformation, such as a linear function $g_l(x) = \boldsymbol{W}_l x$, its spectral norm is $\|g_l\|_{\text{Lip}} = \delta(\boldsymbol{W})$. When the activation function $a_l$ has Lipschitz norm equal to 1 (such as ReLU (Jarrett et al., 2009) and Leaky ReLU (Maas et al., 2013)), we have the following inequality: $\|g_1 \circ g_2\|_{\text{Lip}} \le \|g_1\|_{\text{Lip}}\|g_2\|_{\text{Lip}}$. With this inequality, we obtain the following bound:

$$\|g\|_{\text{Lip}} = \|g_1 \circ a_1 \circ g_2 \circ a_2 \ldots \circ g_d\|_{\text{Lip}} \le \|g_1\|_{\text{Lip}}\|a_1\|_{\text{Lip}}\|g_2\|_{\text{Lip}}\|a_2\|_{\text{Lip}} \ldots \|g_d\|_{\text{Lip}} \quad (5)$$
$$= \Pi_{l=1}^L \|g_l\|_{\text{Lips}} = \Pi_{l=1}^L \delta(\boldsymbol{W}_l)\,.$$

Applying the spectral normalization operation to each weight $\boldsymbol{W}_l$ using $\boldsymbol{W}_{\text{SN}} = \boldsymbol{W}/\delta(\boldsymbol{W})$ enforces $\delta(\boldsymbol{W}_{\text{SN}}) = 1$, so that the right hand side of 5 is above-bounded by 1. This imposes the 1-Lipschitz constraint to the critic function $g$, thus stabilizing the learning signal for training. In practice, the spectral normalization can be estimated efficiently using power iteration.

### 3.3 ACTIVE NEGATIVE-SAMPLE SELECTION (ANS)

Following the discussion in Section 2.2, stable training requires adequate and high-quality negative samples. However, involving too many negative samples far apart from their positive counterparts does not yield effective training, and wastes computational resources. Inspired by the Triplet loss in deep metric learning (Wu et al., 2017; Tschannen et al., 2020), we propose to actively select a relatively small set of negative examples that are most challenging to the discriminator at the current step, thus enabling the model to effectively extract features to distinguish the positives from the negatives.

Specifically, we use two memory banks $\mathcal{B}_u$ and $\mathcal{B}_v$, which store all previously extracted features $\boldsymbol{u}$ and $\boldsymbol{v}$ from seen training instances, respectively. When processing with each new training instances, we actively select the top $K$ nearest neighbors $\mathbf{U}_{\text{nn}} \subset \mathcal{B}_u$ and $\mathbf{V}_{\text{nn}} \subset \mathcal{B}_v$ for $\boldsymbol{u}$ and $\boldsymbol{v}$ via cosine distance. With the QuickSelect algorithm (Hoare, 1961), we are able to identify the top-$K$ negative samples with time complexity $\mathcal{O}(KN)$. Under this setup, 3 can be written as:

$$\mathcal{L}_{\text{ANS}} = \mathbb{E}_{\pi(\boldsymbol{u},\boldsymbol{v})}[\log(g(\boldsymbol{u},\boldsymbol{v}) - \frac{1}{2}\sum_{\boldsymbol{v}'\in\mathbf{V}_{nn}}(\log(1 - g(\boldsymbol{u},\boldsymbol{v}'))) - \frac{1}{2}\sum_{\boldsymbol{u}'\in\mathbf{U}_{nn}}(\log(1 - g(\boldsymbol{u}',\boldsymbol{v})))] \quad (6)$$

When the dataset is large, this can still cost much time in feature searching. Asymmetric Locality Sensitivity Hashing (ALSH) (Shrivastava & Li, 2014) can be applied to hash the representations in a proximity-relevant manner. This helps to reduce the time complexity of ANS to sub-linear (Shrivastava & Li, 2014). In (Schroff et al., 2015) it was found empirically that relaxing the most challenging negative samples to semi-hard negative samples sometimes leads to better results in supervised tasks like classification. This indicates that an approximate retrieval method like ALSH can still perform well. Our observations in experiments are consistent with these findings.

### 3.4 RECLAIM LEARNING PROCEDURE

**Momentum update** Before training, the two memory banks $\mathcal{B}_{\boldsymbol{u}}, \mathcal{B}_{\boldsymbol{v}}$ are initialized with Gaussian noise: $\{\boldsymbol{u}' \sim \mathcal{N}(0, I) | \forall \boldsymbol{u}' \in \mathcal{B}_{\boldsymbol{u}}\}, \{\boldsymbol{v}' \sim \mathcal{N}(0, I) | \forall \boldsymbol{v}' \in \mathcal{B}_{\boldsymbol{v}}\}$. Naively, one can directly replace old features in memory banks with new processed features corresponding to the same input data. However, the performance of such a solution is suboptimal in practice. This is because the target model may change rapidly during the early training stages; such a practice reduces the consistency of the representations, and results in noisy learning signals (He et al., 2020; Wu et al., 2018).

Therefore, we apply the momentum update to mitigate such an issue. Specifically, assume a seen input instance $\boldsymbol{x}$ reappears. There would be a snapshot feature pair previously stored in the $\mathcal{B}_{\boldsymbol{u}}, \mathcal{B}_{\boldsymbol{v}}$ of this $\boldsymbol{x}$, denoted as $\{\tilde{\boldsymbol{u}}, \tilde{\boldsymbol{v}}\}$. We further denote the newly computed (with the current encoder) feature pair correspond to $\boldsymbol{x}$ as $\{\boldsymbol{u}, \boldsymbol{v}\}$, and let $\rho \in (0, 1]$ be the momentum update parameter. We update the feature pairs in $\mathcal{B}_{\boldsymbol{u}}, \mathcal{B}_{\boldsymbol{v}}$ as:

$$\tilde{\boldsymbol{u}} = (1 - \rho)\boldsymbol{u} + \rho\tilde{\boldsymbol{u}}, \qquad \tilde{\boldsymbol{v}} = (1 - \rho)\boldsymbol{v} + \rho\tilde{\boldsymbol{v}} \tag{7}$$

Note that the features in the memory bank are only updated in this way, and they are detached from the computational graph.

**Joint Objective Optimization** With our ANS module, we can obtain the top-$K$ nearest negative features for both $\boldsymbol{u}$ and $\boldsymbol{v}$, denoted $\{\boldsymbol{u}'_i\}_1^K$ and $\{\boldsymbol{v}'_i\}_1^K$, respectively. To get our CL loss, we apply the Wasserstein-distance-based NCE loss by simply imposing the 1-Lipschitz constraint on critic $g$ in (6). Assuming the task-specific loss is $\mathcal{L}_{\text{task}}$, the total loss therefore is formulated as

$$\mathcal{L}_{\text{total}} = \mathcal{L}_{\text{task}} - \lambda \mathcal{L}_{\text{ANS}}, \tag{8}$$

where $\lambda > 0$ is the hyper-parameter controlling the importance of the NCE loss. By minimizing equation 8, we can improve the training model performance. The detailed training procedure is listed in Algorithm 1.

## 4 RELATED WORK

**Connection to Mutual Information Estimation (MIE):** MIE represents a family of MI-based representation learning methods. Early works on MIE, such as InfoMax (Linsker, 1988; Bell & Sejnowski, 1995), advocate maximizing the MI between the input and output. MINE (Belghazi et al., 2018) uses a bi-directional generative model to estimate the MI via the Donsker-Varadhan representation theorem. Deep InfoMax (Hjelm et al., 2019a) further developed MINE by removing one generator, seeking to maximize MI between local features and global features. Our method is more related to the InfoNCE and NCE objectives introduced in (Oord et al., 2018; Gutmann & Hyvärinen, 2010), which are used to estimate the lower bound of the MI. In the proposed RECLAIM, Wasserstein distance is employed to estimate the distribution gap rather than KL-divergence, because it is more stable and robust in practice.

**Contrastive Learning in NLP:** Contrastive learning has been widely used in NLP (Smith & Eisner, 2005; Collobert et al., 2011; Bordes et al., 2013; Hjelm et al., 2019a; Deng et al., 2020). Broadly, CL methods attempt to differentiate observed data from artificial negative examples, to learn representations. In (Gutmann & Hyvärinen, 2010), the Noise Contrastive Estimation (NCE) metric is leveraged to differentiate the target sample from noise samples, to learn word embeddings. Negative sampling proposed by Mikolov et al. (2013b) is a simplified variation of NCE loss, and it achieves great success in learning embeddings. Our CL approach is different from the above, since ($i$) we incorporate Wasserstein distance in NCE, and ($ii$) the proposed negative-sample-selection strategy is different from all other NCE-related works.

**Connection to deep metric learning:** Triplet loss is a classic approach in deep metric learning, and has already been widely used in retrieval models (Lee et al., 2018) and recommendation systems (Chechik et al., 2010). It minimizes the distance between an anchor point to a positive input, meanwhile maximizing its distance to a negative input. Motivated by the triplet loss, some works enforce constraints on more than one negative examples. For example, PDDM (Huang et al., 2016) and Histogram Loss (Ustinova & Lempitsky, 2016) use quadruplets. Moreover, the $n$-pair loss (Sohn, 2016) and Lifted Structure (Oh Song et al., 2016) define constraints on all images in a batch. In these previous works, researchers have focused on enlarging the batch size, since they are only sampling the negative examples within the batch. Our approach incorporates the advantages of those works,

| Model | CoLA | SST-2 | MRPC | MNLI | WNLI | STS-B | RTE | QQP | QNLI |
|---|---|---|---|---|---|---|---|---|---|
| BERT-base | 55.6 | 91.2 | 89.47/85.29 | 83.60/84.15 | 50.7 | 89.2/88.8 | 60.0 | 90.2/87.0 | 90.2 |
| BERT-RECLAIM | 58.8 | 92.7 | 91.0/87.3 | 84.14/84.41 | 56.33 | 90.1/89.5 | 65.0 | 91.0/87.8 | 90.4 |
| RoBERTa-base | 61.3 | 94.1 | 91.1/87.7 | 87.4/87.2 | 53.6 | 90.5/90.3 | 70.8 | 91.3/88.4 | 92.4 |
| RoBERTa-RECLAIM | 63.8 | 94.4 | 93.0/90.2 | 87.7/87.6 | 56.33 | 90.7/90.6 | 74.1 | 91.7/88.9 | 92.9 |

Table 1: Supervised results on GLUE dataset. Darker color indicates more improvement. Better in color.

and moves beyond them to allow active sampling of the most challenging negative pairs from all seen instances within the memory banks. Such a global sampling strategy ensures the selected negative pairs for the same positive pair to be more consistent during training, so that the training can be more stable.

## 5 EXPERIMENTS

Three experiments are conducted to evaluate our approach: ($i$) supervised and semi-supervised natural language understanding (NLU) tasks on the GLUE dataset via BERT (Devlin et al., 2018) and RoBERTa (Liu et al., 2019); ($ii$) image-text cross-domain retrieval task, and ($iii$) text generation task with the VAE framework (Bowman et al., 2016). We set $\lambda = 0.1$ in (6) across all experiment, which are tested on two NVIDIA TITAN X GPUs.

### 5.1 GLUE DATASET CLASSIFICATION

**Dataset:** The General Language Understanding Evaluation (GLUE) benchmark (Wang et al., 2018) is a collection of 9 datasets for evaluating natural language understanding models. Six of these tasks are either single sentence classification or paired sentence classification tasks.

**Implementation details:** We develop our approach on the open-sourced Hugging face Transformers codebase(Wolf et al., 2019)[1]. The hyper-parameters settings, *i.e.*, learning rate, batch size, epoch numbers, etc., are all set to the default setup recommended by the official Transformer repository, for fair comparisons and reproducibility. We report results on the development sets after fine-tuning the pre-trained models (BERT (Devlin et al., 2018) and RoBERTa (Liu et al., 2019)) on each downstream task training data (no ensemble model or multi-task training is used). We utilize the 12-layer architecture (base model) for both BERT and RoBERTa, due to limited computational resources. Parameters $u, v$ are set to be the classification token embedding (CLS) and the embedded input text representation in BERT/RoBERTa, respectively.

We present comparisons on fully supervised tasks in Table 1, with $K = 100$ for RECLAIM. According to this table, RECLAIM consistently improves the performance on all GLUE datasets. For the semi-supervised experiments, we randomly draw $\{1\%, 5\%, 10\%, 20\%, 30\%, 50\%\}$ from each training dataset in GLUE, and results are provided in Table 2. Note that in this case, we only use top-10 ($K = 10$) negative examples, since the training dataset size is largely reduced. It can be seen from Table 2 that our approach generally achieves better results than just fine-tuning BERT on a small dataset.

We note that both fully-supervised and semi-supervised tasks on GLUE do not only serve as classic natural language inference (NLI) problems, but also serve as a testbed for the generalization ability of a representation learner to new tasks. Presumably, results in Tables 1 and 2 suggest that the large-scale pre-trained BERT knowledge has been better transferred to each task in GLUE in our CL approach, as improving the Wasserstein dependency measure (Ozair et al., 2019) can be seen as encouraging the model to be as "lossless" as possible.

### 5.2 TEXT-IMAGE RETRIEVAL

**Dataset:** We evaluate the proposed RECLAIM on both the Flickr30K (Plummer et al., 2015) and MS-COCO (Lin et al., 2014b) datasets. Flickr30K contains 31,000 images, and each photo has five captions. The Flickr30K dataset is split following the same setup as (Karpathy & Fei-Fei, 2015;

---

[1]`https://github.com/huggingface/transformers`, version 2.5.1 using Pytorch 1.2.0 (Paszke et al., 2017)

| dataset | 1% | 5% | 10% | 20% | 30% | 50% |
|---|---|---|---|---|---|---|
| CoLA-BERT | 3.25 | 16.01 | 33.99 | 40.99 | 45.82 | 50.64 |
| CoLA-RECLAIM | 15.11 | 30.78 | 36.80 | 43.74 | 47.23 | 53.64 |
| SST-BERT | 68.53 | 89.15 | 89.80 | 90.32 | 91.25 | 91.80 |
| SST-RECLAIM | 83.28 | 89.25 | 89.90 | 90.41 | 91.79 | 91.92 |
| STSB-BERT | 56.57/43.87 | 56.66/52.27 | 74.64/74.47 | 83.20/83.36 | 85.22/85.41 | 87.34/87.73 |
| STSB-RECLAIM | 56.78/53.42 | 64.11/62.09 | 79.34/79.36 | 83.48/83.28 | 85.33/85.60 | 87.73/87.95 |
| MNLI-BERT | 48.40 | 73.67 | 76.46 | 79.56 | 80.56 | 81.99 |
| MNLI-RECLAIM | 54.22 | 73.89 | 77.07 | 79.63 | 80.65 | 82.59 |

Table 2: Semi-supervised results on GLUE dataset (CoLA, SST-2, STS-B, MNLI). Results for QNLI, MRPC, and QQP are in the Supplementary Material (SM). Darker color indicates more improvement. Better in color.

Faghri et al., 2018). We have a training dataset with 29,000 images, a validation set with 1,000 images, and a test dataset with 1,000 images. MS-COCO contains 123,287 images, with 5 human annotated descriptions per image. We use the same split as in (Faghri et al., 2018), *i.e.*, the dataset is split into 113,287 training images, 5,000 validation images, and 5,000 test images.

**Implementation details.** For this image-text matching task, we use the Adam optimizer (Kingma & Ba, 2015) to train the models. Note that we developed our approach upon SCAN (Lee et al., 2018)[2], by substituting the triplet loss with our RECLAIM. In this experiment, $u$ is the textual feature extracted from a GRU, and $v$ is the image feature extracted from a ResNet. Training details are provided in the Appendix.

The performance of sentence retrieval with image query or image retrieval with sentence query is measured by *recall at* $T$ (R@T) (Karpathy & Fei-Fei, 2015), defined as the percentage of queries that retrieve the correct objects within those with top $T$ highest similarity scores as determined by the model. In this experiment, we can further improve the image-text retrieval results with the basic model design of SCAN. The improvement indicates that features extracted by our CL approach can better capture the common salient information between text and image pairs.

| | Sentence Retrieval | | | Image Retrieval | | | |
|---|---|---|---|---|---|---|---|
| Method | R@1 | R@5 | R@10 | R@1 | R@5 | R@10 | Rsum |
| DPC (ResNet) Zheng et al. (2017) | 55.6 | 81.9 | 89.5 | 39.1 | 69.2 | 80.9 | 416.2 |
| SCO (ResNet) Huang et al. (2018) | 55.5 | 82.0 | 89.3 | 41.1 | 70.5 | 80.1 | 418.5 |
| SCAN (Faster R-CNN, ResNet) Lee et al. (2018) | 67.7 | 88.9 | 94.0 | 44.0 | 74.2 | 82.6 | 452.2 |
| Vanilla CL | 68.2 | 89.3 | 94.7 | 47.4 | 75.2 | 82.9 | 457.7 |
| **RECLAIM** | **69.5** | **91.2** | **95.5** | **48.6** | **77.1** | **84.2** | **466.1** |
| DPC (ResNet) Zheng et al. (2017) | 41.2 | 70.5 | 81.1 | 25.3 | 53.4 | 66.4 | 337.9 |
| SCO (ResNet) Huang et al. (2018) | 42.8 | 72.3 | 83.0 | 33.1 | 62.9 | 75.5 | 369.6 |
| SCAN (Faster R-CNN, ResNet)Lee et al. (2018) | 46.4 | 77.4 | 87.2 | 34.4 | 63.7 | 75.7 | 384.8 |
| Vanilla CL | 47.1 | 79.6 | 88.1 | 35.5 | 64.2 | 76.9 | 391.4 |
| **RECLAIM** | **49.7** | **81.0** | **89.8** | **37.9** | **66.7** | **78.2** | **403.3** |

Table 3: Image-Text retrieval results with Recall@$K$ (R@K). Upper panel: Flickr30K, lower panel: MSCOCO.

## 5.3 TEXT GENERATION

**Dataset:** We further evaluate our model on an unsupervised text generation task. Two commonly-used datasets are employed for this task, the Penn Treebank (PTB) (Marcus et al., 1993; Bowman et al., 2016) and the Yelp corpora (Yang et al., 2017; He et al., 2019). PTB is a relatively small dataset with sentences of varying lengths, whereas Yelp contains larger amounts of long sentences. Detailed statistics of these two datasets are summarized in the Appendix.

**Implementation details:** To ensure a fair comparison and reproducibility between models, we develop our model based on an existing codebase[3]. The encoder and decoder are both 1-layer LSTMs, and the hyper-parameter setup follows the instructions within the original codebase. In this task, the $u$ is the latent code in text VAE and $v$ is the word embedding vectors of the input text. The most commonly used metrics are applied to evaluate the learned language model, as listed in Table 4.

According to Table 4, by simply adding our proposed CL method, the base model can be further improved in terms of most of the automatic metrics. Lower negative ELBO indicates our approach

---

[2]https://github.com/kuanghuei/SCAN
[3]https://github.com/fangleai/Implicit-LVM

yields a better language model. Larger KL divergence and larger Active Unit (AU) (Burda et al., 2015) indicate that the latent space is more sufficiently made use of. We also observed that the posterior-collapsing problem is alleviated, with improved mutual information and KL (Fang et al., 2019); this is presumably due to the fact that we add additional CL objective to the latent code and output to improve the MI between them.

| Methods | -ELBO↓ | PPL↓ | KL↑ | MI↑ | AU↑ | Methods | -ELBO↓ | PPL↓ | KL↑ | MI↑ | AU↑ |
|---|---|---|---|---|---|---|---|---|---|---|---|
| VAE | 102.6 | 108.26 | 1.08 | 0.8 | 2 | VAE | 357.9 | 40.56 | 0.0 | 0.0 | 0 |
| $\beta$(0.5)-VAE | 104.5 | 117.92 | **7.50** | 3.1 | 5 | $\beta$(0.4)-VAE | 358.2 | 40.69 | 4.2 | 2.0 | 4 |
| SA-VAE | 102.6 | 107.71 | 1.23 | 0.7 | 2 | SA-VAE | 355.9 | 39.73 | 2.8 | 1.7 | 8 |
| Cyc-VAE | 103.1 | 110.50 | 3.48 | 1.8 | 5 | Cyc-VAE | 355.9 | 39.73 | 3.8 | 2.4 | 11 |
| iVAE | 87.6 | 54.46 | 6.32 | 3.5 | 32 | iVAE | 348.2 | 36.70 | 7.6 | **4.6** | 32 |
| iVAE-vanilla CL | 88.1 | 55.2 | 1.6 | 4.1 | 32 | iVAE-vanilla CL | 350.1 | 37.1 | 6.6 | 3.7 | 32 |
| iVAE-RECLAIM | **85.6** | **48.7** | 5.88 | **4.43** | **32** | iVAE-RECLAIM | **346.6** | **36.5** | 4.2 | 4.4 | **32** |

Table 4: Variational language modeling results on PTB (left) and Yelp dataset (right).

## 5.4 ABLATION STUDY

**Choice of $K$**  We seek to further investigate how the negative sample size $K$ influences the effectiveness of our model. To this end, we choose different $K = \{1, 10, 100, 300\}$ in ANS for comparison. Besides testing different $K$s with ANS, we also test an alternative approach, where we random sample 80% of features from the memory bank as negative samples instead of applying ANS. We denote this method simply as the 80% Method. Also, the in-batch method denotes that we only use negative samples within the batch. The results can be found in Table 5. Note that those two tricks can be viewed as two different ways for constructing vanilla contrastive learning (Vanilla CL) algorithm.

| Model | CoLA | SST-2 | MRPC | MNLI | WNLI | STS-B | RTE | QQP | QNLI |
|---|---|---|---|---|---|---|---|---|---|
| 80% w/o ANS | 57.8 | 92.1 | 89.7/85.7 | 83.84/84.21 | 56.33 | 90.1/89.5 | 63.1 | 91.0/87.8 | 90.2 |
| In-batch | 55.5 | 91.0 | 89.48/85.30 | 83.60/84.11 | 51.3 | 88.9/87.8 | 61.3 | 90.0/86.6 | 88.9 |
| RECLAIM K=1 | 56.6 | 91.6 | 89.43/85.16 | 83.44/84.01 | 53.7 | 88.7/87.4 | 62.1 | 89.8/86.3 | 89.7 |
| RECLAIM K=10 | 57.8 | 92.1 | 90.2/86.3 | 83.71/84.12 | 55.42 | 89.4/89.2 | 64.5 | 90.4/87.3 | 90.3 |
| RECLAIM K=100 | **58.8** | 92.7 | 91.0/87.3 | 84.14/84.41 | **56.33** | **90.1/89.5** | **65.0** | 91.0/87.8 | 90.4 |
| RECLAIM K=300 | 58.4 | **92.7** | **91.1/87.5** | **84.20/84.52** | 55.63 | **90.1/89.5** | 64.6 | **91.2/87.9** | **90.5** |

Table 5: Results for different K choices on GLUE dataset. Th 80% method are also listed as a comparison.

From Table 5 we observe that when $K$ is small, the improvement is limited. In some tasks, such as MNLI, it is even worse than the BERT-base model. This finding is consistent with arguments from previous works (Wu et al., 2017; Tschannen et al., 2020). For $K = 100$ and $K = 300$, comparable results are often observed, and either of them outperforms the others on certain tasks; $K = 300$ seems to work better on tasks with larger datasets such as MNLI/QNLI. We hypothesize that this is because larger datasets contain more high-quality contrastive examples than smaller datasets, thus allow using a large $K$ without introducing much noise. Both of them show better results than the 80% Method without ANS.

**Computational Efficiency**  MNLI, the biggest dataset in GLUE, is employed as a running-time benchmark to evaluate the computational efficiency among different approaches. We record the training time for the original BERT, $K = 100$, and the 80% Method. Without any contrastive regularization, BERT takes approximately 45 minutes per epoch. For $K = 100$, RECLAIM needs 47 minutes per epoch. The 80% Method takes 81 minutes per epoch. The memory usage for BERT-base is around 7.5GB for a batchsize (per GPU) of TITAN X. $K = 100$ takes an additional 200MB, and the 80% Method takes full 12GB memory capacity. These empirical findings provide evidence that our method can be both efficient and effective. Due to space limitations, other ablation studies, including the investigation of different $\lambda$ choices, are provided in the Appendix.

## 6 CONCLUSIONS

We have proposed a novel contrastive learning (CL) framework, RECLAIM, for natural language processing tasks. Our approach improves the "contrast" in the feature space, in an attempt to make the features learned by the representation encoder to be more distinguishable, representative and informative. We identified the challenges in CL training and proposed several remedies to alleviate

these issues. Extensive experiments show that consistent improvements over a variety of NLP tasks, demonstrating the effectiveness of our approach.

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

| dataset | 1% | 5% | 10% | 20% | 30% | 50% |
|---|---|---|---|---|---|---|
| QNLI-BERT | 72.69 | 83.75 | 85.83 | 87.82 | 88.93 | 89.96 |
| QNLI-ours | 72.52 | 83.96 | 85.44 | 88.03 | 88.67 | 90.02 |
| MRPC-BERT | 31.62/0.00 | 68.38/81.22 | 69.40/80.07 | 71.85/80.47 | 75.65/83.65 | 84.07/88.95 |
| MRPC-ours | 31.62/0.00 | 68.38/81.22 | 70.59/80.38 | 72.88/80.80 | 79.82/86.46 | 81.92/87.65 |
| QQP-BERT | 75.43/70.95 | 82.98/78.16 | 84.65/79.09 | 86.76/81.90 | 87.82/83.77 | 89.19/85.51 |
| QQP-ours | 73.56/70.33 | 82.57/78.45 | 84.91/80.14 | 87.23/83.22 | 88.33/84.49 | 89.59/85.94 |

Table 6: Semi-supervised results on GLUE dataset (QNLI, MRPC, and QQP).

## A  ALGORITHM

Here is the detailed algorithm for our RECLAIM framework.

---
**Algorithm 1** RECLAIM training procedure.

---
1: **Input:** batch size $n$, dataset $\mathbf{X}$, momentum parameter $\eta$, maximum number of iterations $N$.
2: **for** itr $= 1, \ldots N$ **do**
3:     Sample input $\boldsymbol{x} \sim \mathbf{X}$ and initialize two memory bank $\mathcal{B}_{\boldsymbol{u}}, \mathcal{B}_{\boldsymbol{v}}$
4:     Get features: $\boldsymbol{u}, \boldsymbol{v} = \text{Network}(\boldsymbol{x})$
5:     Update $\mathcal{B}_{\boldsymbol{u}}, \mathcal{B}_{\boldsymbol{v}}$ by using Equation 7
6:     Using ANS algorithm to draw $K$ negative features for $\boldsymbol{u}$ and $\boldsymbol{v}$:
            $\{\boldsymbol{u}'_i\}_1^K \sim \mathcal{B}_{\boldsymbol{u}}, \quad \{\boldsymbol{v}'_i\}_1^K \sim \mathcal{B}_{\boldsymbol{v}}$
7:     Stop gradient for $\{\boldsymbol{u}'_i\}_1^K, \{\boldsymbol{v}'_i\}_1^K, \mathcal{B}_{\boldsymbol{u}}, \mathcal{B}_{\boldsymbol{v}}$
8:     Calculate Equation 6 with Wasserstein constrained critic $g$
9:     Minimize the Ensemble loss: $\mathcal{L}_{\text{total}} = \mathcal{L}_{\text{task}} - \lambda \mathcal{L}_{\text{ANS}}$
10: **end for**

---

## B  MORE RESULTS

### B.1  MORE RESULTS FOR SEMI-SUPERVISED EXPERIMENT

Results for QQP, MRPC and QNLI are shown here in 6.

### B.2  MORE VAE RESULTS

| Methods | -ELBO↓ | PPL↓ | KL↑ | MI↑ | AU↑ | Methods | -ELBO↓ | PPL↓ | KL↑ | MI↑ | AU↑ |
|---|---|---|---|---|---|---|---|---|---|---|---|
| VAE | 102.6 | 108.26 | 1.08 | 0.8 | 2 | VAE | 357.9 | 40.56 | 0.0 | 0.0 | 0 |
| VAE-Reclaim | 102.1 | 107.15 | 1.2 | 0.9 5 | | VAE-Reclaim | 352.1 | 40.2 | 2.1 | 1.7 | 10 |
| Cyc-VAE | 103.1 | 110.50 | 3.48 | 1.8 5 | | Cyc-VAE | 355.9 | 39.73 | 3.8 | 2.4 | 11 |
| Cyc-VAE with | Reclaim 98.8 | 105.7 | 4.18 | 2.2 | 10 | Cyc-VAE-Reclaim | 350.2 | 39.73 | 4.5 | 3.3 | 25 |

Table 7: Variational language modeling results on PTB (left) and Yelp dataset (right).

### B.3  ABLATION STUDY

**Importance of the Wasserstein regularizer**

| Model | CoLA | SST-2 | MRPC | MNLI | WNLI | STS-B | RTE | QQP | QNLI |
|---|---|---|---|---|---|---|---|---|---|
| BERT-base Devlin et al. (2018) | 55.6 | 91.2 | 89.47/85.29 | 83.60/84.15 | 50.7 | 89.2/88.8 | 60.0 | 90.2/87.0 | 90.2 |
| BERT-CL | 57.8 | 92.1 | 89.7/85.7 | 83.84/84.21 | 56.33 | 90.1/89.5 | 63.1 | 91.0/87.8 | 90.2 |
| BERT-CL+Wasserstein only | 58.1 | 92.3 | 89.8/86.2 | 84.1/84.25 | 56.13 | 90.1/89.5 | 64.7 | 91.0/87.8 | 90.3 |
| BERT-RECLAIM | 58.8 | 92.7 | 91.0/87.3 | 84.14/84.41 | 56.33 | 90.1/89.5 | 65.0 | 91.0/87.8 | 90.4 |

Table 8: Results on GLUE dataset.

| Model | CoLA | SST-2 | MRPC | MNLI | WNLI | STS-B | RTE | QQP | QNLI |
|---|---|---|---|---|---|---|---|---|---|
| BERT-base Devlin et al. (2018) | 55.6 | 91.2 | 89.47/85.29 | 83.60/84.15 | 50.7 | 89.2/88.8 | 60.0 | 90.2/87.0 | 90.2 |
| RECLAIM $\lambda = 0.01$ | 55.8 | 91.5 | 89.66/85.38 | 83.64/84.18 | 52.7 | 89.4/89.0 | 63.0 | 90.6/87.3 | 90.2 |
| RECLAIM $\lambda = 0.1$ | 58.8 | 92.7 | 91.0/87.3 | 84.14/84.41 | 56.33 | 90.1/89.5 | 65.0 | 91.0/87.8 | 90.4 |
| RECLAIM $\lambda = 0.5$ | 57.1 | 92.2 | 90.2/86.8 | 83.81/84.31 | 55.7 | 89.8/89.2 | 65.0 | 90.8/87.4 | 90.3 |
| RECLAIM $\lambda = 1.0$ | 55.5 | 90.8 | 89.33/85.25 | 83.39/83.38 | 51.7 | 89.2/88.1 | 62.1 | 90.0/86.6 | 90.0 |

Table 9: Results for different $\lambda$ choices on GLUE dataset.

| Model | CoLA | SST-2 | MRPC | MNLI | WNLI | STS-B | RTE | QQP | QNLI |
|---|---|---|---|---|---|---|---|---|---|
| BERT-large | 62.8 | 94.2 | 91.9/88.7 | 87.6/87.3 | 55.6 | 90.5/90.4 | 72.3 | 91.3/88.4 | 92.5 |
| RECLAIM | 63.8 | 94.4 | 93.0/90.2 | 87.7/87.6 | 56.33 | 90.7/90.6 | 74.1 | 91.7/88.9 | 92.9 |

Table 10: Supervised results on GLUE dataset with BERT-large model.

**importance of $\lambda$ choices** We also test the effect of choosing different $\lambda \in \{0.01, 0.1, 0.5, 1\}$. As shown in Table 9, We can see that when $\lambda \leq 0.5$, RECLAIM can consistently outperform the BERT-base model. It may because we need to re-scale the contrastive loss to the same numerical scale as the task-specific loss.

**Experiment on BERT-Large** We also test GLUE experiment on BERT-large model, to see whether our proposed algorithm can still be effective. Results can be found in Table 10.

## B.4 VQA TEST

We also tested our approach on Visual Question Answering (VQA) 2.0 task Goyal et al. (2017), which contains human-annotated QA pairs on COCO images (Lin et al., 2014a). For each image, an average of 3 questions are collected, with 10 candidate answers per question. The most frequent answer from the annotators is selected as the correct answer.

Following previous work Kim et al. (2018), we take the answers that appear more than 9 times in the training set as candidate answers, which results in 3129 candidates. Classification accuracy is used as the evaluation metric, defined as $\min(1, \frac{\text{\# humans provided ans.}}{3})$.

In this setup, we choose $u, v$ as question features and image features respectively. By applying our RECLAIM approach directly to BAN model, we can see a improvement over the VQA task as shown in Table 2.

| Method | VQA-score |
|---|---|
| BAN (Kim et al., 2018) | 66.06 |
| Ours | 66.36 |

Table 11: VQA validation dataset results

## C TRAINING DETAILS

**Image-Text Retrieval** For the Flickr30K data, we train the model for 30 epochs. The initial learning rate is set to 0.0002, and decays by a factor of 10 after 15 epochs. For MS-COCO data, we train the model for 20 epochs. The initial learning rate is set to 0.0005, and decays by 10 after 10 epochs. We set the batch size to 128, and threshold the maximum gradient norm to 2.0 for gradient clipping. We also set the dimension of the GRU and joint embedding space to 1024, and the dimension of the word embedding to 300.

**GLUE** We choose batch size as 32 for all 9 GLUE tasks, and $2 \times 10^{-5}$ is the starting learning rate. For each task, we only perform 3 epochs, since some datasets, such as RTE is quite small, they can be easily got over-fitted.

**dimension of features $u$ and $v$** Note that our CLAIM formulation do not require $u$ and $v$ to have a matching dimension. Contrastive learning seeks to compare $\pi(u, v)$ to $\pi(u)\pi(v)$, not comparing $u$ directly with $v$. Though in practice, we often map $u$ and $v$ to matching dimensions via MLP or RNN to balance their respective contribution to the loss.

**Architecture for g:** We use two different MLP layers first to map $u, v$ into the same dimension, and then we feed both into a three layer MLP. Details will be included in our next revision. For instance, in BERT model, we will map word representations and hidden states to dim $= 64$.

**choice of $u$ and $v$** : In GLUE experiment: $u$ is chosen as word vectors (from one-hot tokens to real vectors), $v$ is chosen as the BERT output features with dim=768.

In VAE setup: $u$ is chosen as word vectors (from one-hot tokens to real vectors), $v$ is chosen as the encoded latent variable $z$.

In Image-Text retrieval setup: $u$ is chosen as word vectors (from one-hot tokens to real vectors), $v$ is chosen as the image features.

