# OpenReview forum: "Towards Robust and Efficient Contrastive Textual Representation Learning"
_ICLR.cc/2021/Conference — Reject_

### Official Review · AnonReviewer1 · 2020-10-13
**Sound empirical results, unclear about goal of paper (representation learning vs regularization)**

**Rating:** 6
**Confidence:** 3

**Review:**

-----------------------------------------
Summary
-----------------------------------------
This paper proposes an approach to improve (supervised and unsupervised) representation learning for text using constrastive learning. The proposed approach augments standard contrastive learning with: (1) Spectral-norm regularization of the critic to estimate the Wasserstein distance instead of the KL (as in the Wasserstein GAN-style approach), (2) Active negative sampling to select hard negative examples, and (3) momentum-based updates of intermediate features. The resulting contrastive learning objective can be combined with standard supervised and unsupervised objectives to improve downstream tasks.

None of these techniques is novel on its own to my knowledge, so the main contribution here is to show that combining these techniques for text representation learning yields strong empirical results. This combined approach, called RECLAIM, is applied to a suite of standard NLP tasks: transfer learning, image-text retrieval, and language modeling, where it is found to outperform standard baselines.

-----------------------------------------
Strengths
-----------------------------------------
- Sound improvements on top of existing finetuning methods for GLUE  (though I have questions about the semi-supervised setup; see questions).

- Evaluation across a wide range of setups (though language modeling results aren't really valid in my opinion due to the iVAE perplexity numbers being not exact; see weaknesses).

- Ablation studies across various augmentations, though it would have been interesting to also ablate on Wasserstein vs KL as well.

-----------------------------------------
Weaknesses
-----------------------------------------
- My biggest gripe is that the paper is presented as a "representation learning" work, yet much of the focus is on performance on underlying tasks. That is, the contrastive learning objective is simply used as an additional regularizer on top of existing objectives. Implicit in this is the idea that "good representations should lead to good downstream performance", but it would have been great to do closer analyses of the learned representations to see how this contrastive learning objective results in different representations (see some example analyses that could be conducted in the questions section).

- Given that RECLAIM is really a source of regulariazation, I would have liked to seen it compared to existing regularization methods such as MixOut on GLUE (https://arxiv.org/abs/1909.11299).

- Results on language modeling are not really valid since the iVAE codebase (on which this work is based) does not actually calculate perplexity (they only estimate perplexity, which is a drawback of the iVAE work). The language modeling results would be much more compelling if the authors showed improvements on top of works that calculate perplexity more properly, e.g. with a variational lower bound or an importance-sampled estimate (e.g. as done in works such as Cyc-VAE).

- It seems like this objective can be combined with self-supervision (e.g. masked LM objective) to aid in better BERT-like models (which would have been a very interesting result), but this seems to have been left unexplored.

-----------------------------------------
Questions/Comments
-----------------------------------------
- How do representations learned this way differ from the standard approach? There are many interesting analyses that could be conducted for the different tasks. For example: on GLUE, which layers of BERT are finetuned more (or less) with this method; on image-text retrieval, what are some mistakes made by the standard approach that this method is able to fix?; on language modeling, how do the learned latent codes differ compared to iVAE (i.e. is it more sensitive to nouns/verbs, etc?)

- Does the semi-supervised experiments utilize the full unlabeled data? (i.e. do you train with the contrastive objective on unlabeled data, and the supervised + contrastive objective on the labeled data?)

- Does the switch from KL to Wasserstein distance make the lower bound on MI (equation 4) invalid?

- The results on GLUE would be much easier to read if the average results across tasks are also presented
---------
After rebuttal: Thank you for the response. I have decided to leave my score unchanged.

---

> ### Author Response · Authors · 2020-11-20
> **Response to AnonReviewer1**
>
> Thanks very much for your insightful comments!
>
> closer analysis: Yes, we totally agree with AnonReviewer1. That's why we conducted a semi-supervised learning for representation testing. We will add more analysis in our next revision, i.e. showing the latent code in VAE learned via Reclaim, and visualize the learned text/image feature pairs via Reclaim.
>
> Compared with Mixout: We will cite and discuss the Mixout paper. We will try to implement Mixout in our framework, and make comparisons in our next version. However, we think our method is orthogonal to the method proposed in Mixout in fine-tuning BERT, since Mixout is trying to regularize BERT training with dropout and dropconnect within the model architecture, but in our work, we propose a new loss function to regularize the training. We can even combine both Mixout with our framework, this will be our future work.
>
> Results on IVAE: We totally agree with R1 that iVAE is not calculating the actual PPL. We will updated the results in the appendix. Here are the VAE and Cyc-VAE improvement results:
>
> PTB:
>
> VAE 102.6 108.26 1.08 0.8 2
>
> VAE-Reclaim 102.1 107.15 1.2 0.9 5
>
> Cyc-VAE 103.1 110.50 3.48 1.8 5
>
> Cyc-VAE with Reclaim 98.8. 105.7 4.18 2.2 10
>
> Yelp:
>
> VAE 357.9 40.56 0.0 0.0 0
>
> VAE-Reclaim 352.1 40.2 2.1 1.7 10
>
> Cyc-VAE 355.9 39.73 3.8 2.4 11
>
> Cyc-VAE-Reclaim 350.2 39.73 4.5 3.3 25
>
> Unexplored: Thanks for pointing out this! We totally agree that the best way to show the capability of our proposed method is to train a BERT-like model with MLM + our loss objective from scratch. We are currently working on this however we are not able to include the results by the time of submission, We will add this in our future version.
>
> For the comments & questions:
>
> 1. Thanks very much for this advice! We will try to add these analyses (refer to the first bullet "closer analysis").
>
> 2. Yes, we do utilize the full unlabeled data.
>
> 3. This can still hold. As we discussed in sec 2.2, Eqn (4) did not require the extra constraint on function g. All we did in our framework is just enforcing extra smoothness to function g.
>
> 4. Thanks for this advice. We will fix this in the next revision.

---

### Official Review · AnonReviewer3 · 2020-10-28
**Contrastive learning for NLP with Wasserstein constraints and fast/dynamic selection of high-quality negative samples**

**Rating:** 6
**Confidence:** 4

**Review:**

This paper presents a new contrastive learning methods based on intermediate textual feature pairs, which stabilizes training and overcomes vanishing contrasting signals by imposing Wasserstein constraints on the critic via spectral regularization, and leverages a fast and dynamic selection of high-quality negative samples.

Overall, this is an interesting work on Contrastive learning for NLP, which
-	switches from the MI-based NCE to a Wasserstein-based NCE for more robust and stable training
-	creates a fast (with sub-linear Asymmetric Locality Sensitivity Hashing) and dynamically (with momentum update) select the most challenging negative examples on-the-fly.
-	seems to get solid performance gains on wide range of downstream applications.

My major concern is about the clarity of the paper and some detail information of model and experimental design.  It seems not very easy to reproduce corresponding codes and experiments only based on the current paper content. At least, there is quite some space to improve readability. For instance:
-	The concrete u and v in each experiment should be more clearly explained, especially for u.
-	Detail design and Implementation of g
-	"random sample 80% of features" seems not clear enough for understanding
-	More explanation for “Stop gradient for…” in Algorithm 1
-	Organic integration of section 3.2/3.3/3.4 based on holistic and detail explanation of Figure 1
-	More clear illustrations on unsupervised/supervised/semi-supervised experimental designs and processes

Minor issues:
-	Some notations are not self-contained. For example, “sup_x” in section 3.2 is not defined.
-	Reference section should be more carefully checked. For instance, Hjelm et al. 2019a and 2019b are pointing to the same paper, and Lagler et al. 2013 seems to point to a wrong GPT2 paper which is not in the NLP domain.

---

> ### Author Response · Authors · 2020-11-20
> **Response to AnonReviewer3**
>
> Thanks very much for your comments!
>
> The concrete of u and v: The choices for u and v are detailed listed in the last section of our appendix. We will try to elaborate this in the main draft in our next version.
>
> Implementation of g: we have discussed the implementation of g in the appendix (last paragraph in Section C).
>
> "random sample 80% of features": For the vanilla contrastive learning method, we need to sample a large size of negative examples from the memory bank. Suppose the memory bank shares the same size (N) as the dataset, we will randomly sample 0.8N negative examples from the memory bank for contrastive learning.
> We will make this statement clearer.
>
> Stop gradient in Algorithm 1: since the negative examples are sampled from the memory bank, we do not want the training gradient flowing back to the memory bank to update the negative examples.
> Because if we do let the negative examples be updated, then the training process will not be as smooth as our setups (similar reason as why choosing momentum update).
> We will include a more detailed explanation in our next version.
>
> Organic integration & More clear illustrations:
> We have a framework overview in Section 3.1, briefly introducing how different parts work in our framework. Thanks for this advice, we will try to update 3.1 to make it more clearer to understand.
>
> Minor issues:
> Thanks very much for pointing out these issues! We will fix them in the next revision.

---

### Official Review · AnonReviewer4 · 2020-10-29
**Limited novelty, unfair comparisons**

**Rating:** 3
**Confidence:** 3

**Review:**

The paper argues that there are some challenges in learning text representations with vanilla contrastive learning techniques, due to the metric-aware property of  KL divergence as well as the difficulties in selecting good negative examples. It then presents two remedies: employ a Wasserstein constraint to the critic function and a simple active sampling strategy for negative examples.

I think the paper is not strong enough to get into ICLR. Firstly, the Wasserstein constraint has been proposed to improve general representation learning in Ozair et al. (2020). The novelty of this work is very limited compared to Ozair et al. (2020). The active sampling contribution is a simple heuristic. Secondly, the authors only compare the proposed method with the non-CL baseline. A vanilla CL baseline should be included in all the experiments. Finally, the improvement margins are also pretty small.

The writing can be much better if the authors can use examples for concrete tasks. For instance, I am not sure the distinction between u and v representations after reading sec. 2.2.

---

> ### Author Response · Authors · 2020-11-20
> **Response to AnonReviewer4**
>
> Thanks very much for your valuable time!
>
> The novelty of this work: We agree that our work is inspired by Ozair et al. (2020), however, the formulations in that paper are totally different from ours. The Wasserstein distance we used in the paper is trying to enforce extra smoothness in the critic function g. Our work can still be viewed as a new framework to estimate the mutual information. But Ozair et al. (2020) estimate the Wasserstein discrepancy instead of KL, and Eqn 4 may not hold any more in their setup.
> And Ozair et al. (2020) did not propose a better way for negative sampling, which is also very crucial in CL method. Since many works did not even alter the basic formulation of CL, but mainly focus on how to build a good negative example sampling strategy, for instance, SimCLR, MOCO.
> This is consistent with our observation, because if we directly apply the method from Ozair et al. (2020), there is no difference in model performance between Ozair and vanilla CL (BERT experiment).
> Also, the ANS method is not just a heuristic invention. As we discussed in the third paragraph in Section 4 ("Connection to deep metric learning"), we try to combine some well-known techniques in deep metric learning to contrastive learning algorithms for better performance.
>
> Not comparing with CL baselines: In the ablation study, we have done the comparison with vanilla CL methods with our framework. The vanilla CL methods include “randomly sample large number of negative examples”-based CL and “in-batch negative example”-based CL.
> Also, we have updated the CL baseline results for other experiments (please check our response to AnonReviewer2).
>
> What are u and v: in section 2.2, we are trying to give a general concept of how to build contrastive learning in our framework. And we have discussed how to choose u and v for different experiment setups in Section 5, also with more details in the appendix. We will make it more clearer in our next version.
>
> Improvement is pretty small: As we showed the additional results in the appendix, a 12-layer BERT model with our new objective function is comparable to the 24-layer BERT model. That means our framework can help to reduce the computational burden from training a larger model.
> Also, we tested our framework on VAEs and Image-Text retrieval, the improvements are not marginal at all.

---

### Official Review · AnonReviewer2 · 2020-11-03

**Rating:** 5
**Confidence:** 3

**Review:**

The paper aims to improve contrastive learning for text representation by tackling two challenges: (1) The conventional mutual-information (MI) based objective can results in unstable training due to the sensitivity of KL to changes of representation; (2) contrastive learning could require a large number of negative samples, which is inefficient. For (1), the paper replaces the KL loss with Wasserstain distance; while for (2), the paper selects top-K most difficult negative samples (defined as the nearest neiborhors to the current representations).

The contrastive learning loss is then used to augment the standard task-specific loss to enhance learning and enable semi-supervised training. Experiments on benchmarks for text classification (GLUE), retrieval, and language modeling show the proposed method improve over standard supervised/semi-supervised learning.

The proposed method is intuitive and simple, largely stitching together previous successful techniques (e.g., Wasserstein distance for robustness, hard negative samples, momentum update).
Section 2.2 gives a nice summary of several core challenges of contrastive learning.

Questions / weaknesses:

1) In section 2.2, is it necessary to introduce v? -- Constrative learning is usually seen as maximizing the MI between input w and representation u, i.e., MI(w, u). Why here MI(v, u), e.g., Eq.(4), is used?

2) I'm a bit surprised that the experiments do not include any comparison with previous contrastive learning methods, but only compared with models without contrastive learning.
Similarly, the ablation study only compares results of negative sample sizes (K). It's necessary to include more comparsions to show how much improvement each of the two proposed techniques (Wasserstein distance, hard negative samples) reaches.

3) The caption of Table.2 says "Results for QNLI, MRPC, and QQP are in the Supplementary Material (SM). ". Yet those results are not included in the suppementary.

---

> ### Author Response · Authors · 2020-11-20
> **Response to AnonReviewer2**
>
> Thanks very much for your comments!
>
> Why introduce v: admittedly, contrastive learning usually seeks to maximize the mutual information between input and feature. The reason we introduced u and v is because we want to have a more general form, where u and v can go beyond the conventional setup as input and feature. Specifically, we believe the contrastive learning can be applied to pairs of representations regardless of whether or not they are input-feature pairs, as long as the paired representations manifest certain characteristics of the data.  In our setup, the u-v pairs can be either input-feature pairs, or the feature-output pairs. Different choices are made based on different experiment goals. We will make this clearer.
>
> Not comparing with base CL method: We have tested the vanilla CL method for GLUE dataset, the test results are the top two lines in Table 5:
> “80% w/o ANS” is the baseline method, also is the most basic CL method, which will draw a large number of negative examples from the entire memory bank . “In-batch” is also a CL method baseline, instead of sampling negative examples from the memory bank, it just samples negative examples within the batch.
> Therefore, our ablation study (Table 5) not just focuses on the sampling size of K, but also considers (1) whether ANS is better than other negative sampling tricks (comparing with CL baselines), (2) which K best works for ANS.
>
> To test the importance of Wasserstein distance, we will add new ablation study in the Appendix in the next version.
> We also list our new Results here:
>
> Testing the importance of Wasserstein regularization:
>
> BERT-base 55.6 91.2 89.47/85.29 83.60/84.15 50.7 89.2/88.8 60.0 90.2/87.0 90.2
>
> BERT-CL 57.8 92.1 89.7/85.7 83.84/84.21 56.33 90.1/89.5 63.1 91.0/87.8 90.2
>
> BERT-CL+Wasserstein only 58.1 92.3 89.8/86.2 84.1/84.25 56.13 90.1/89.5 64.7 91.0/87.8 90.3
>
> BERT-RECLAIM 58.8 92.7 91.0/87.3 84.14/84.41 56.33 90.1/89.5 65.0 91.0/87.8 90.4
>
> Other experiment with CL baseline method:
> 1. Image-text retrieval:
> Flickr30K
>
> SCAN  67.7 88.9 94.0 44.0 74.2 82.6 452.2
>
> Vanilla CL: 68.2 89.3 94.7 47.4 75.2 82.9 457.7
>
> RECLAIM 69.5 91.2 95.5 48.6 77.1 84.2 466.1
>
> MSCOCO
>
> SCAN 46.4 77.4 87.2 34.4 63.7 75.7 384.8
>
> Vanilla CL 47.1 79.6 88.1 35.5 64.2 76.9 391.4
>
> RECLAIM 49.7 81.0 89.8 37.9 66.7 78.2 403.3
>
> 2. VAEs:
> PTB:
> iVAE 87.6 54.46 6.32 3.5 32
>
> iVAE-vanilla CL: 88.1 55.2 1.6 4.1 32
>
> iVAE-RECLAIM 85.6 48.7 5.88 4.43 32
>
> Yelp:
>
> iVAE 348.2 36.70 7.6 4.6 32
>
> iVAE-vanilla CL 350.1 37.1 6.6 3.7 32
>
> iVAE-RECLAIM 346.6 36.5 4.2 4.4 32
>
>
> Not include results: Thanks for pointing this out, we will update those testing results in our appendix.

---

### Decision · Program_Chairs · 2021-01-07
**Final Decision**

**Decision:**

Reject

**Comment:**

This work brings improvement to contrastive learning method for text data by
combining a Wasserstein objective with a "memory bank" strategy
for getting (and updating) hard negative samples.
The approach leads to small but consistent improvements across a variety of representation learning tasks in both supervised and unsupervised settings. While the paper makes a useful contribution and evaluates with some success on downstream tasks, the reviewers would like to see some intrinsic, qualitative discussion of the representation learning itself, and in comparison to more powerful contrastive learning methods. Overall the work falls below the acceptance threshold in a very competitive venue, so I cannot recommend acceptance.

Even after discarding one uninformative review, the consensus remains borderline. The reviewers are, however, appreciative of the additional clarifying experiments provided by the authors. In the internal discussion, a concern was raised that the BERT baselines may not have a fair chance to compete, as they are not fine-tuned on the unsupervised data in the same way that the proposed method is, leading to possibly overestimating the improvement.  I encourage the authors to consider this.

Finally, while some concerns about clarity have been addressed, some remain in the current version. In particular, I spot at least three duplicate and inconsistent entries, for Hjelm et al, Lin et al, and McAllester and Stratos.)